# The Stable Fly (*Stomoxys calcitrans*) as a Possible Vector Transmitting Pathogens in Austrian Pig Farms

**DOI:** 10.3390/microorganisms8101476

**Published:** 2020-09-25

**Authors:** Lukas Schwarz, Andreas Strauss, Igor Loncaric, Joachim Spergser, Angelika Auer, Till Rümenapf, Andrea Ladinig

**Affiliations:** 1Department for Farm Animals and Veterinary Public Health, University Clinic for Swine, University of Veterinary Medicine, 1210 Vienna, Austria; andyst@gmx.at (A.S.); andrea.ladinig@vetmeduni.ac.at (A.L.); 2Department of Pathobiology, Institute of Microbiology, University of Veterinary Medicine, 1210 Vienna, Austria; igor.loncaric@vetmeduni.ac.at (I.L.); joachim.spergser@vetmeduni.ac.at (J.S.); 3Department of Pathobiology, Institute of Virology, University of Veterinary Medicine, 1210 Vienna, Austria; angelika.auer@vetmeduni.ac.at (A.A.); till.ruemenapf@vetmeduni.ac.at (T.R.)

**Keywords:** hemotrophic mycoplasmas, PRRSV, PCV2, bacteria

## Abstract

This pilot study aimed to investigate stable flies from Austrian pig farms for the presence of defined swine pathogens, such as porcine reproductive and respiratory syndrome virus (PRRSV), porcine circovirus 2 (PCV2), hemotrophic mycoplasmas in ingested blood and/or body parts and bacteria on the surface of the flies. Furthermore, the use of stable flies as a diagnostic matrix for the detection of pathogens in the ingested pig blood should be investigated. In total, 69 different microorganisms could be found on the surface of tested *S. calcitrans* from 20 different pig farms. *Escherichia coli* was the most common bacterium and could be found on flies from seven farms. In seven farms, hemotrophic mycoplasmas were detected in stable flies. PRRSV could not be found in any of the samples of these 20 farms but PCV2 was detected in six farms. Whether the stable fly can be used as a matrix to monitor the health status cannot be accurately determined through this study, especially in regard to PRRSV. Nevertheless, it might be possible to use the stable fly as diagnostic material for defined pathogens like *Mycoplasma suis* and PCV2.

## 1. Introduction

The stable fly, *S. calcitrans*, is often found in swine farms. Its significance in food animal production can be divided into direct and indirect influences. Direct influences can include restlessness, pain due to biting, stress, loss of blood, reduced feed intake, lesions of the skin followed by local inflammation and immunosuppression, while indirect effects are due to the transmission of infectious agents [1]. This is supported by Zumpt (1973), who mentioned that the main harming effects caused by stable flies are (i) biting dependent disturbance, (ii) loss of blood and toxic reactions caused by *Stomoxys* saliva and (iii) the transmission of pathogens [2].

Stable flies feed on blood from their hosts one or two times per day. Usually, they do not finish blood feeding on one animal, since the host animals try to defend against the painful fly bites. Additionally, stable flies can be disturbed by other flies during blood feeding. Both causes for interruption of blood feeding end up in a change of biting site or change of the host [3,4]. *S. calcitrans* can ingest 3.2–22.6 mg blood [5]. Mellor et al. (1987) investigated the survival of African Swine Fever Virus (ASFV) in stable flies after feeding on infected pigs or after feeding on viremic blood in the laboratory. In this study, it was shown that ASFV can survive or stay infectious in stable flies for up to 48 h [6]. Although the regurgitation of ingested blood by stable flies seems improbable [3], the ingestion of stable flies which fed on ASFV viremic pigs by non-infected pigs was at least a possible way of ASFV transmission under experimental conditions [7]. For other host species, such as horses, stable flies are relevant vectors for transmission of infectious diseases [8,9,10,11], but in pigs only a few studies exist [1,6,12].

The objective of this study was to investigate the role of stable flies as a possible vector for swine pathogens under conditions found in Austrian farms. Furthermore, we tried to clarify if *S. calcitrans* can be used as a diagnostic matrix for the surveillance of defined pathogens in farms where stable flies occur.

## 2. Materials and Methods

As this investigation was a pilot study, we did not calculate any sample size. Stable fly specimens were collected from a total of 20 Austrian pig farms either during routine farm visits or during consultation farm visits performed by the University Clinic for Swine as support for the corresponding herd health veterinarian. Both clinically unsuspicious farms and farms with current health/disease problems were chosen (Table 1).

### 2.1. Sampling

Stable fly specimens were collected between June 2018 and July 2019. Stable flies were mainly collected in the gestation area, as there the most specimens were observed. Fly specimens were collected from different animals. In each farm, a total of 40 stable flies were caught by hand wearing sterile gloves. Collected flies were placed in 15-mL plastic tubes and kept on crushed ice for transportation to the laboratory and for immobilization. Ten of these 40 fly specimens were used for bacteriological examination. For the virological investigation, another 20 flies were prepared and the remaining 10 flies were stored at −20 °C for possible additional or repeated investigations.

### 2.2. Processing of Flies for Further Analysis

For further processing, flies were kept in a refrigerator to cool down to a temperature of about 4 °C. In this immobilised stage all flies were prepared for investigations on frozen thermal packs under a laminar flow hood to guarantee a stage of insensitiveness during preparation [12,13].

### 2.3. Microbiological Examination

For bacteriological examination a pool sample of 10 stable flies was washed in 1 mL physiologic sodium chloride (Kochsalz “Braun” 0,9%—Infusionslösung^®^, B. Braun Melsungen AG, Melsungen, Germany) solution on a vortex shaker (2 min, 4 °C) to get a suspension containing bacteria from the fly surface [14]. After mixing, the suspension was transferred to a 1.5 mL Eppendorf tube and sent to the Institute of Microbiology, University of Veterinary Medicine Vienna, for isolation of cultivable bacteria.

For cultivation of bacteria a loop of the washing solution (approximately 10µL) was streaked onto BD Columbia Blood Agar with 5% *v*/*v* sheep blood and incubated in aerobic/microaerophilic (5% CO_2_)/anaerobic conditions, Columbia CNA Improved II agar with 5% *v*/*v* sheep blood and BD MacConkey II(all from Becton Dickinson, Germany). Agar plates were incubated at 37 °C for 24–48 h. Isolation of fungi was conducted on BD Sabouraud Agar with gentamicin and chloramphenicol (Becton Dickinson, Germany) at 28 °C for 3–4 days. After incubation, one colony representing a distinct colony morphotype was identified to species level (if applicable) by matrix-assisted laser desorption ionization–time of flight mass spectrometry (MALDI-TOF MS) (Bruker Daltonik, Germany).

The remaining flies, which had been washed, were used for the detection of mycoplasmas by PCR employing universal primers (hmyc-forward: 5′-GGCCCATATTCCTACGGGAAGCAGCAGT-3′, hmyc-reverse: 5′-TAGTTTGACGGGCGGTGTGTACAAGACCTG-3′) for the amplification of the 16S rRNA gene of hemoplasmas and closely related non-hemotrophic mycoplasma species as previously described [15]. Resulting amplicons were sequenced at LGC Genomics, Berlin, Germany, and sequences compared to entries in the GenBank nucleotide database using BLASTN algorithm.

### 2.4. Virological Investigations

All preparation steps were performed on frozen thermal packs (−20 °C). For dissecting, the flies were put into a sterile plastic bag onto the thermal pack to keep the flies immobilised and insensitive during the preparation. For virological investigations the mouth parts (proboscis) and the abdominal parts of 20 flies were prepared and pooled using a scalpel [3]. Fly samples were homogenized in 1 mL PBS and four sterile 3-mm stainless steel beads in an automatic TissueLyser II (Qiagen, Hilden, Germany) followed by centrifugation at 16,000× *g* for one minute. Next, 140 µL of fly lysate was used for nucleic acid extraction, which was performed on a QIAcube instrument using the QIAamp Viral RNA Mini QIAcube Kit (Qiagen, Hilden, Germany) according to the manufacturer’s instructions. Negative controls (PBS only) were prepared side by side with each extraction. PCV-2 specific nucleic acids were detected using forward primer (pCV2-2.1F) 5′-GAGTCTGGTGACCGTTGCA-3′, reverse primer (pCV2-2R) 5′-YCCCGCTCACTTTCAAAAGTTC-3′ and probe 5′-FAM-CCCTGTAACGTTTGTCAGAAATTTCCGCG-BHQ1-3′ as described by Bernd Hoffmann (VIR02-pCV-2-qPCR-FLI). The PCR was carried out using the Luna^®^ Universal Probe qPCR Master Mix (New England Biolabs, Germany) according to the manufacturer’s instructions on a Rotor-Gene Q 5plex (Qiagen, Hilden, Germany). For absolute quantification, a serial dilution of a defined DNA plasmid standard was run in parallel with all samples. Results were given in genome equivalents (GE)/mL fly lysate, positive qPCR results <1 × 10^4^ GE/mL were reported as “below level of quantification”. For detection of PRRSV-1 specific nucleic acids an about 400 base pairs long fragment of N gene was amplified using forward primer (PRS133) 5′-ATGGCCAGCCAGTCAATC-3′ and reverse primer (PRS 134) 5′-TCGCCCTAATTGAATAGGTG-3′ [16]. The RT-PCR was carried out using OneTaq One-Step RT-PCR Kit (New England Biolabs, Germany). After two pre-steps (48 °C for 20 min and 94 °C for 1 min), the PCR was run for 45 cycles of denaturation at 94 °C for 15 s, primer annealing at 54 °C for 30 s and primer extension at 68 °C for 30 s. Final extension was at 68 °C for 5 min. RT-PCR products were visualized using the capillary electrophoresis device QIAxcel Advanced System (Qiagen, Hilden, Germany).

Samples for virological investigations were limited to PRRSV and PCV2 as both are prevalent in Austria and regularly cause viremia in infected animals [17,18]. As Austria is free of Aujeszky’s disease and African swine fever, we did not investigate those notifiable diseases.

## 3. Results

### 3.1. Microbiological Examination

In total, 69 isolates of microorganisms were identified (Table 2) on the surface of the investigated stable flies from the 20 different farms. These isolates consisted of 37 different species. Almost all isolates were at least evaluated at the genus level, except one isolate which could not be identified using MALDI-TOF analysis. Furthermore, one out of 69 isolates could be classified as a member of yeast, whereas the remaining ones were all bacterial species.

The most frequently recovered species was *Escherichia* (*E.*) *coli*, which was isolated from stable flies from 7/20 farms (35%). Other frequently detected species were *Aerococcus viridans* (6/20, 30%), *Enterobacter cloacae* (5/20, 25%), *Morganella morganii* (5/20, 25%) and *Staphylococcus xylosus* (4/20, 20%). In total, eight different staphylococcal species could be identified. If all staphylococcal isolates were grouped together, 25% (17/68) of all bacterial isolates were members of the genus *Staphylococcus* (Table 3). Enterobacteria were found more frequently (39.7%, 27/68). Table 3 provides an overview of all detected isolates with their frequencies of detection.

In total, on all farms, a minimum of one species and a maximum of nine different species were detected, which resulted in an average of 3.45 different bacterial species on stable flies per farm.

Porcine hemoplasmas (*Mycoplasma suis*, *Mycoplasma parvum*) were found in stable flies from 7 farms (35%) (Table 2).

### 3.2. Virological Investigations

None of the pooled samples (proboscis and abdomen from 20 specimens) used for virological examination was positive for PRRSV RNA by RT-PCR (Table 2). However, the quantitative PCR (qPCR) for detection of PCV2 DNA was positive in 6 out of the 20 farms (Table 2). Two of these positive samples could be quantified (1.63 × 10^4^ GE/mL and 1.65 × 10^4^ GE/mL). Both quantifiable samples originated from farms with a history of porcine reproductive and respiratory syndrome. The remaining four positive samples were all below the limit of quantification (<1 × 10^4^ GE/mL fly lysate) and originated from a farm with weak born suckling piglets and increased return-to-estrus-rates (*n* = 1), a farm with confirmed PCV2-reproductive disease (*n* = 1), a farm with a PRRS outbreak/eperythrozoonosis (*n* = 1), and a farm with tail biting (*n* = 1).

## 4. Discussion

The stable fly may have negative effects on pig production not only by inducing stress due to biting acts, but also by acting as a vector for pathogens [19,20]. Flies, such as the common housefly, *Musca domestica*, in general play an important role in transmitting bacteria and resistances against antimicrobials, as they contribute to the active circulation and maintenance of resistant bacteria in farm animal production [21,22]. Compared to the literature, we found similar results, as several different bacterial species were found on Austrian stable flies too probably indicating a similar role in transmission of resistances against antimicrobials. As we did not investigate for antimicrobial resistance, this is a limiting factor not only of our study, but also of other studies, which did not publish any susceptibility testing data [23,24,25]. Therefore, future research to obtain data on the antimicrobial resistance of bacteria from stable flies is needed. Antimicrobial resistance needs to be monitored in the near future.

In two previous studies, a maximum of 33 different bacterial species were isolated from stable flies [1,23]. In our study, we detected a total of 36 different species. Although some bacteria may be natural colonizers of stable flies and may not be pathogens of pigs, the potential of *S. calcitrans* serving as a vector for a relatively broad spectrum of cultivable bacterial species, which also may carry resistance genes, has to be considered in combating antimicrobial resistance in bacteria. Assuming an important role of stable flies in the transmission and distribution of antimicrobial resistance, reducing antimicrobial resistance must also include effective insect management in farms to prevent spreading of potential resistant bacterial strains not only from animals to animals, but also from animals to humans [26]. Stable flies may carry bacteria not only on their body surface but also on their mouthparts and may directly transmit bacteria with their proboscis during the blood-feeding act.

In a Brazilian study, *E. coli* comprised 31% in the family *Enterobacteriaceae* isolated from *S. calcitrans* [24]. Similarly, in our study 25% of *Enterobacteriaceae* were *E. coli*. As *Enterobacteriaceae* form an important family which contribute to the distribution of resistance genes via plasmid exchange [27], the relatively high isolation rate of enterobacteria from stable flies should be seriously considered when developing strategies for reducing antimicrobial resistance in bacteria. Besides *E. coli* we detected some other species of the order *Enterobacterales* such as *Morganella morganii*, *Proteus* sp., *Enterobacter cloacae* and *Citrobacter feundii.* While one may discuss the importance of enterobacteria in clinical disease and distribution of antimicrobial resistance, studies have shown that enterobacteria are part of the gut microbiome of healthy pigs with *E. coli* being the predominant species [28]. Since we did not analyse antimicrobial susceptibility and virulence genes, we can only speculate about the actual relevance and importance of the isolated enterobacteria in our study. Interestingly, *Staphylococcus hyicus,* a common and widely distributed skin pathogen and skin commensal of pigs, was not detected in any of the analysed stable flies. This may be because at the time point of sampling none of the pigs showed clinical signs of exudative epidermitis, neither its local nor generalized form. Additionally, we could not detect any streptococcal species from stable fly specimens. This is supported by results of a Brazilian study in which no streptococci were detected [23]. However, in synanthropic flies (also *S. calcitrans*) collected in Germany at different sites (dog pound, poultry house, cattle barn, horse stable, and pig pen) streptococci were recovered. Unfortunately, it cannot be concluded from this study whether these streptococci were isolated from stable flies or from other fly species [25]. Furthermore, streptococci are natural colonizers of the porcine skin, as was shown in Danish slaughter pigs [29]. One may have expected at least one or the other streptococcal isolate, because during blood feeding the flies have direct contact to skin microbiota either with their proboscis or with their legs. As we were not able to find any indications that *S. calcitrans* is colonized with streptococci, we hypothesize that it plays only a minor role in streptococcal infections, such as *Streptococcus suis* or *Streptococcus dysgalactiae* infections.

Stable flies may transmit bacteria not only via their body surface, but also during the blood feeding act. It is known that *S. calcitrans* serves as a potential mechanical vector of *Mycoplasma suis* under natural conditions, but with limited effectiveness [12]. However, in our study it was possible to detect hemotrophic mycoplasmas in stable flies from 7/20 (35%) farms. One farm had a longer history of eperythrozoonosis and despite the lack of clinical symptoms such as anaemia, jaundice, fever, and growth retardation at the time point of sampling, it was possible to detect hemotrophic mycoplasmas. Whether or not these hemotrophic mycoplasmas were infectious, cannot be investigated properly since bioassays would be required to test their infectiousness.

A study investigating *S. calcitrans* as a possible vector of PRRSV in swine found it was not possible to transmit PRRSV between two animals after blood feeding. However, the possibility of contamination of the uncontaminated environment of another farm by stable flies which fed on viremic pigs could not be excluded [13], as stable flies are known to travel for long distances of more than one kilometre [30]. A limiting factor of that study was that the authors did not investigate whether the virus load of the blood meal was sufficient to cause infection in pigs by contacting a contaminated environment. Another explanation may be a possible low sensitivity of RT-PCR used for detection. In our study we could not detect any positive PRRSV genome fragments in flies, despite the PRRS-positive herd status of some farms (5/20). On the one hand that may be due to the sensitivity of the RT-PCR protocol used. On the other hand, it might be explained by low to non-existent viral loads in the blood of the pigs. It is possible that none of the investigated farms were sampled at the time point of the acute outbreak/virus circulation at which many highly viremic animals occur. Moreover, all PRRSV positive farms used modified live-virus vaccines at the time point of sampling. Therefore, a limited number of viremic pigs may have been present, which may also have had low virus titres in serum, depending on the circulating PRRSV strain. A limiting factor of our study was the fact that *S. calcitrans* could mainly be observed and caught in the gestation area or breeding unit of a farm. Stable flies need continuous breeding environments for their larvae to successfully establish a stable fly population in a farm. Gestation areas and breeding centres, at least in Austria, hardly ever get cleaned and disinfected thoroughly due to the continuous use, so that over a quite long time period stable fly maggots can successfully develop to imagos. In contrast, farrowing units, nursery, weaning and fattening units are used in an all-in/all-out production system, which allows thorough cleaning and disinfection. As a consequence, stable fly maggots lack a consistent and sustainable habitat or get killed by disinfectants. As sows after PRRSV infection are usually viremic for only a few days [31], this may explain the negative test results of our study. Furthermore, another study showed that in 4.5-month-old experimentally infected pigs, initial viremia lasted 7 days and thereafter the virus could only be detected irregularly until day 18 after infection [32]. This short period of viremia and the fact that ingested blood in sampled stable flies gets reduced by the digestive tract of the fly until the flies, which are still alive, have been cooled down for preparation [3], may be other explanations for the negative test results. Therefore, we are not able to clearly state whether stable flies may be used as a diagnostic matrix for the detection of PRRSV in a farm or not. It should be clearly stated that we investigated stable flies from randomly chosen farms and we did not specifically collect flies from acute PRRS outbreak farms. For investigating the role of stable flies in the diagnostics of PRRS, controlled experimental infection trials would be necessary in future.

In case of PCV2, the causative agent of PCV2-systemic disease, it is supposed that flies in general play an important role as vectors [33]. This assumption cannot be dismissed, as stable flies in 6 out of 20 farms in our study were positive for PCV2 genome fragments. One farm had a confirmed problem with PCV2-reproductive (PCV2-RD) disease in sows with the consequence of stillborn and/or mummified foetuses. Stable flies from that farm tested positive. One may speculate, that *S. calcitrans* plays a substantial role in the transmission of PCV2 from viremic piglets or sows to naïve sows with the consequence of PCV2-RD. In another farm which reported stillborn piglets from time to time, stable flies tested positive for PCV2 DNA using qPCR. Despite no systematic prevalence studies on the use of vaccination of sows against PCV2 currently existing in Austria, we know from our own experience and from reports from the field that it is rather unusual in Austria to vaccinate sows against PCV2. We speculate that heterogenous and/or instable immunity levels against PCV2 in sow herds are important risk factors, which offer PCV2 the opportunity to actively infect susceptible sows and circulate within the breeding sows causing weak born, stillborn or mummified piglets. For the latter farm the responsible herd veterinarian decided to reintroduce vaccination of sows against PCV2 during the suckling period. Half a year later and after consequent vaccination of all sows, no further cases of stillborn piglets were reported. Consequently, testing of stable flies in breeding areas and or gestation units may help detecting subclinical infections of sows or can at least enhance the probability of detecting active virus circulation in the sow herd. Moreover, testing stable flies for the presence of PCV2 genome is a cost effective and gentle tool regarding animal welfare replacing blood sampling. One limitation may be that no data is available for the interpretation of quantified PCV2 qPCR results. However, it seems that PCV2 viral load in *S. calcitrans* is far away from the values for PCV2-associated diseases, where moderate to high amounts of PCV2 DNA genome copies should be detected to talk of clinical disease [34]. Nonetheless, stable flies which tested positive for PCV2 DNA at least originated from farms with a clinical history, which may also be linked to PCV2 infections, either subclinical or clinical. We hypothesize that any positive PCR reaction, regardless of the quantity of detected virus copies, may be a sign of clinical affection of sows in the corresponding herd. One may speculate that only stable flies of farms with PCV2-associated diseases will result in positive qPCR-reactions which of course has to be confirmed in future studies.

## 5. Conclusions

To conclude, stable flies can act as natural vectors for different pathogenic microorganisms in swine [1,6,12], whereas it is just assumed that they serve as vectors for some other pathogens [1,3,13]. The results of our study strongly suggest that stable flies in Austrian pig farms are carriers of several different bacterial species and may have also served as vectors for PCV2 and hemotrophic mycoplasmas. Although we cannot evaluate the explicit role of *S. calcitrans* as a vector, we think that the role of stable flies in disease and pathogen transmission in Austria is underestimated. However, it seems that stable flies may be used as a diagnostic matrix for the detection of PCV2 and hemotrophic mycoplasmas. It would have been beneficial for our study to explicitly collect stable flies from PRRSV-positive farms with an acute outbreak (clinical symptoms) to verify if stable flies can be used as a diagnostic matrix for testing against PRRSV. Indeed, this would need further research with a focus on the comparison with gold standard techniques for the corresponding pathogen. The use of stable flies would be beneficial, because it is a pool sample of 10-20 flies which fed on different sows. Furthermore, using stable flies for herd health surveillance would be a cost-effective and humane alternative to blood sampling or other invasive sampling techniques in swine and especially in sows.

## Figures and Tables

**Table 1 microorganisms-08-01476-t001:** Overview of the study farms and their clinical status. In total, pigs of five farms were clinically unsuspicious and did not show any clinical symptoms at the time point of sampling. All farms vaccinated piglets against PCV2.

Farm	Date	Case History/Clinical Symptoms	PRRS-Status
1	06/2018	Infertile sows	Positive, vaccinated
2	06/2018	Clinically unsuspicious	Negative, non-vaccinated
3	07/2018	Increased return-to-oestrus rate, vaginal discharge	Unknown, non-vaccinated
4	07/2018	Post-partum dysgalactia	Negative, non-vaccinated
5	09/2018	Clinically unsuspicious	Negative, non-vaccinated
6	09/2018	Enzootic pneumonia	Negative, non-vaccinated
7	09/2018	Suckling piglet diarrhoea due to *Escherichia coli*	Negative, non-vaccinated
8	09/2018	Clinically unsuspicious	Negative, non-vaccinated
9	09/2018	Clinically unsuspicious	Negative, non-vaccinated
10	09/2018	Clinically unsuspicious, PRRS outbreak 10 years ago	Unknown, non-vaccinated
11	01/2019	Increased return-to-oestrus rate, chronic bacterial cystitis	Negative, non-vaccinated
12	02/2019	Vaginal discharge, vaginitis	Positive, vaccinated
13	03/2019	PRRS outbreak and swine dysentery in fattening pigs	Positive, vaccinated
14	03/2019	Increased return-to-oestrus rate, weak born piglets	Negative, non-vaccinated
15	04/2019	PCV2-reproductive disease	Positive, vaccinated
16	04/2019	*Glaesserella parasuis*-related lameness in sows suspected	Negative, non-vaccinated
17	04/2019	PRRS outbreak, eperythrozoonosis	Positive, vaccinated
18	06/2019	PRRS outbreak	Positive, vaccinated
19 **	06/2019	Tail biting in fattening pigs due to ascariasis	Unknown, non-vaccinated
20	07/2019	PRRS unstable farm	Positive, vaccinated

PRRS-status: vaccinated = modified live virus vaccines; negative = clinically unsuspicious for any PRRS-related symptoms and no positive PCR/ELISA-test result over the last year. ** Farm 19 was the only fattening farm and all other farms were either piglet-producing farms or farrow-to-finish farms.

**Table 2 microorganisms-08-01476-t002:** Overview of all results shown on individual farm level.

Farm	Bacteria		Viruses	
	Species	HM	PRRSV	PCV2
1	*Morganella morganii*	neg.	neg.	neg.
	*Bacillus pumilus*			
	*Citrobacter freundii*			
	*Streptomyces* sp. **			
2	*Aerococcus viridans*	neg.	neg.	neg.
	*Pseudomonas fulva*			
	*Staphylococcus xylosus*			
3	anaerobic bacteria *	pos.	neg.	neg.
	*Morganella morganii*			
	*Bacillus* sp. **			
	*Staphylococcus sciuri*			
4	*Arthrobacter* sp. **	neg.	neg.	neg.
	*Enterobacter cloacae*			
	*Escherichia coli*			
	*Aerococcus viridans*			
	*Staphylococcus xylosus*			
5	*Staphylococcus sciuri*	pos.	neg.	neg.
	*Staphylococcus xylosus*			
	*Aerococcus viridans*			
	*Pantoea calida*			
	*Acinetobacter* sp. **			
6	*Staph. epidermidis*	neg.	neg.	neg.
	*Staphylococcus xylosus*			
	*Raoultella terrigena*			
	*Morganella morganii*			
	*Enterobacter cloacae*			
7	*Candida* sp. **	neg.	neg.	neg.
	*Staph. saprophyticus*			
	*Escherichia coli*			
	*Proteus mirabilis*			
	*Lactococcus lactis*			
	*Aeromonas caviae*			
	*Enterobacter cloacae*			
	*Klebsiella pneumoniae*			
8	*Morganella morganii*	neg.	neg.	neg.
	*Staphylococcus chromogenes*			
9	*Escherichia coli*	pos.	neg.	neg.
	*Klebsiella pneumoniae*			
	*Myroides* sp. **			
	*Enterobacter cloacae*			
10	*Escherichia coli*	pos.	neg.	neg.
	*Clostridium butyricum*			
	*Aerococcus viridans*			
	*Alcaligenes faecalis*			
11	*Staphylococcus haemolyticus*	neg.	neg.	neg.
	*Aerococcus viridans*			
12	*Staphylococcus equorum*	neg.	neg.	neg.
	*Arthrobacter* sp. **			
13	*Staphylococcus equorum*	neg.	neg.	neg.
	*Aerococcus viridans*			
	*Staphylococcus chromogenes*			
14	*Escherichia coli*	neg.	neg.	pos., <LOQ
	*Enterococcus faecalis*			
	*Staphylococcus chromogenes*			
15	*Clostridium perfringens*	neg.	neg.	pos., <LOQ
	*Providencia rettgeri*			
16	*Clostridium perfringens*	neg.	neg.	neg.
	*Escherichia coli*			
	lactose-neg. *Escherichia coli*			
17	CN *Staphylococcus* sp. **	pos.	neg.	pos., <LOQ
	*Escherichia coli*			
	*Providencia rettgeri*			
18	*Enterococcus faecalis*	neg.	neg.	1.65 × 10^4^ GE/mL
	*Myroides odoratimimus*			
	*Providencia alcalifaciens*			
19	*Staphylococcus succinus*	pos.	neg.	pos., <LOQ
20	*Staphylococcus sciuri*	pos.	neg.	1.63 × 10^4^ GE/mL
	*Morganella morganii*			

CN = coagulase-negative; neg. = negative; pos. = positive; GE = genome equivalents; LOQ = limit of quantification; mL = millilitre; HM = hemotrophic mycoplasma; PRRSV = Porcine Reproductive and Respiratory Syndrome Virus; PCV2 = Porcine Circovirus 2; * This isolate could not be further characterised at a species level. ** These isolates were not further analysable using MALDI-TOF.

**Table 3 microorganisms-08-01476-t003:** Detected bacterial isolates and the fungus isolate from *S. calcitrans* body surface. The frequency was calculated from the total number of all bacterial isolates (*n* = 68).

Species	*N*	Frequency of Bacterial Isolates in %
*Escherichia coli*	7	10.29
*Aerococcus viridans*	6	8.82
*Morganella morganii*	5	7.35
*Staphylococcus xylosus*	4	5.88
*Enterobacter cloacae*	4	5.88
*Staphylococcus sciuri*	3	4.41
*Staphylococcus chromogenes*	3	4.41
*Staphylococcus equorum*	2	2.94
*Providencia rettgeri*	2	2.94
*Klebsiella pneumoniae*	2	2.94
*Enterococcus faecalis*	2	2.94
*Clostridium perfringens*	2	2.94
*Arthrobacter* sp. **	2	2.94
*Streptomyces* sp. **	1	1.47
*Staphylococcus succinus*	1	1.47
*Staphylococcus saprophyticus*	1	1.47
*Staphylococcus haemolyticus*	1	1.47
*Staphylococcus epidermidis*	1	1.47
*Raoultella terrigena*	1	1.47
*Pseudomonas fulva*	1	1.47
*Providencia alcalifaciens*	1	1.47
*Proteus mirabilis*	1	1.47
*Pantoea calida*	1	1.47
*Myroides* sp. **	1	1.47
*Myroides odoratimimus*	1	1.47
lactose-negative *Escherichia coli*	1	1.47
*Lactococcus lactis*	1	1.47
coagulase-negative *Staphylococcus* sp. **	1	1.47
*Clostridium butyricum*	1	1.47
*Citrobacter freundii*	1	1.47
*Candida* sp. **	1	1.47
*Bacillus* sp. **	1	1.47
*Bacillus pumilus*	1	1.47
anaerobic bacteria *	1	1.47
*Alcaligenes faecalis*	1	1.47
*Aeromonas caviae*	1	1.47
*Acinetobacter* sp. **	1	1.47

* This isolate could not be further characterised at a species level. ** These isolates were not further analysable using MALDI-TOF.

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
