# Peer review of "The Stable Fly (Stomoxys calcitrans) as a Possible Vector Transmitting Pathogens in Austrian Pig Farms"

_microorganisms, 2020, doi:10.3390/microorganisms8101476_

Round 1
Reviewer 1 Report
I consider this to be an introductory paper to determine the incidence of potential pathogen transmission by stable flies in swine facilities in Austria. The work was quite thorough and did give reason for needed continuation of this line of work with additional focus on the more important aspects. I rated this as a paper of high interest because stable fly is not usually incriminated in pathogen transfer and because swine health through successful disease management is critical. Should the fly sample size be larger initially to determine the size needed to produce the necessary results?
The main problem see with this paper is the English. The topic is quite difficult for an English speaker to word craft, which makes it even more tricky for non-native speakers. I made suggested changes for some sentences so please check to be sure that my suggestions did not change the intended meaning. In some cases word choice could be improved. Please try to have this read over by a native English speaker because much is lost by the style and sentence structure as it stands now.
The results could be more strongly stated in some places. This could have been also a result of the English. I hope the paper can be revised to improve it. The work is very interesting and timely. One other aspect of stable fly biology that was not mentioned is the stable fly's ability to disperse from one location to another. These flies can travel 8 km per hour and in many cases adult populations on one farm were produced on another farm several km away. See the old reference from the DDR: Steinbrink, H. 1989. On the distribution of Stomoxys calcitrans (Diptera: Muscidae) in stables. Angew. Parasit. 30: 57-61. [in German]. More recent publications from Germany or western Europe are few.
The reviewed paper is attached.

Author Response
Dear Reviewer!
Many thanks for your review and for your recommendations and corrections, which we highly appreciate. We considered a native speaker who proof read the manuscript and gave input to improve the English style. In the following we will comment point by point on your comments and recommendations.
Line 28: Stomoxys calcitrans and stable fly were removed, as they are not necessary to be mentioned in the key words.
Line 41-42: We followed your corrections.
Line 43: viraemic was changed throughout the manuscript to viremic.
Line 45-52: We followed your corrections.
Line 67-69: We followed your corrections.
Line 74: We added hood.
Line 85: The wrong and additional "and" was deleted.
Line 103: "Next" was added at the beginning of the sentence.
Line 139-146: We followed your corrections.
Line 154: "used" was added.
Line 165: As we used British English, we did not change the wording of housefly.
Line 168: Regarding your comment A4 we added a clarification" Compared to the literature we found similar results, as ....
Line 172-175: We followed your corrections.
Line 175 Comment A5: As we were investigating bacteria at all, we did not change the wording. Pathogens are in included in the term bacteria. But we like to talk of bacteria in general.
Line 179-202: WE followed your corrections.
Regarding your comment on the fly sample size, we cannot change the situation now, as the study is finished now. For future studies it may be considered to increase the fly sample size to answer unsolved questions. As there are only few studies available at the moment, we will try to use this pilote study for propper study planing. Thanks a lot for the valuable comment.
Regarding your comment on the travel habits of stable flies was considered and we added at lien 218 "...could not be excluded [13], as stable flies are known to travel for long distances of more than one kilometre [30] (Steinbrink, 1989)."
Again we want to thank you for your valuable comments and corrections. As an English native speaker proof read the manuscript, we hope that we could improve the style of our manuscript and we hope, that you feel comfortable with it now.
Reviewer 2 Report
Important and well-written paper. I only have a few minor comments and questions, which are made on the attached mark-up of the pdf.
Line 67 - State here which animals were the focus of the collections. Also, was care taken to collect flies from more than one animal per farm? Sometimes there is a temptation to collect many flies from a single heavily-infested animal.
Table 2, Line 134 - Define abbreviations in table footnote
Line 172 - Not sure what is meant by "seems inevitable". I believe that you are saying that it is important/essential/critical to obtain this sort of information and that this should be a priority for future work.
Line 182 - House flies regurgitate crop contents on animal food and the environment, and this is thought to play a role in movement of antibiotic-resistant bacteria. How do you suppose that stable flies might transfer bacteria to host animals?
Line 231 - It seems to me that this is a fact, not a hypothesis
Line 250 - Do you mean that this assumption "is supported by our observation that stable flies in 6 out of 20 farms ...."
Additional minor comments are in the file attached.

Author Response
Dear Reviewer!
Many thanks for your valuable comments, corrections and recommendations. We highly appreciate it and hope that we could improve our manuscript in the revised version. As an English native speaker proof-read the manuscript, some wordings changed slightly.
Line 41-42: This sentence was rewritten: S. calcitrans can ingest 3.2 – 22.6 mg blood [5].
Line 45: We followed your correction.
Line 67: As recommended we added additional information: Stable flies were mainly collected in the gestation area, as there the most specimens were observed. Fly specimens were collected from different animals.
Line 92: Have was changed to "had".
Table 2: Abrreviations are explained now in the table legend.
Line 139: As it was commented by the second reviewer and the native speaker we did not change due to your comment.
Line 142: We used the recommendation and corrections of reviewer 2.
Line 146: We followed your correction.
Line 172: This sentence was rewritten: Therefore, future research to obtain data on the antimicrobial resistance of bacteria from stable flies is needed.
Line 182: We added an explanatory sentence: Stable flies may carry bacteria not only on their body surface but also on their mouthparts and may directly transmit bacteria with their proboscis during the blood-feeding act.
Line 219: This sentence was rewritten due to a recommendation of the native speaker: A limiting factor of that study was that the authors did not investigate whether the virus load of the blood meal was sufficient to cause infection in pigs by contacting a contaminated environment.
Line 231: We reformulated this sentence: Stable flies need continuous breeding environments for their larvae to successfully establish a stable fly population in a farm.
Line 235: We changed imagines to imagos.
Line 237: We followed your recommendation.
Line 250: WE changed debilitated to dismissed.
Line 285: The comma was deleted.
References: We added the missing information regarding the volumes. We want to appologize for that formating mistake. Obviously the citation program did not produce the correct citation style.
Round 2
Reviewer 1 Report
Most if not all of my questions were addressed and sentence structure was greatly improved. I have attached the manuscript because of one and maybe two more very minor sentence changes, but nothing more. I hope some follow up work can be conducted and published.
